

# Enhancing bug allocation in software development: a multi-criteria approach using fuzzy logic and evolutionary algorithms

Chetna Gupta[1,*] and Varun Gupta[2,3,*]

[1] Jaypee Institute of Information Technology, Noida, India
[2] Department of Economics and Business Administration, University of Alcala, Alcalá de Henares, Madrid, Spain
[3] Multidisciplinary Research Centre for Innovations in SMEs (MrciS), GISMA University of Applied Sciences, Potsdam, Germany
* These authors contributed equally to this work.

## ABSTRACT

A bug tracking system (BTS) is a comprehensive data source for data-driven decision-making. Its various bug attributes can identify a BTS with ease. It results in unlabeled, fuzzy, and noisy bug reporting because some of these parameters, including severity and priority, are subjective and are instead chosen by the user's or developer's intuition rather than by adhering to a formal framework. This article proposes a hybrid, multi-criteria fuzzy-based, and multi-objective evolutionary algorithm to automate the bug management approach. The proposed approach, in a novel way, addresses the trade-offs of supporting multi-criteria decision-making to (a) gather decisive and explicit knowledge about bug reports, the developer's current workload and bug priority, (b) build metrics for computing the developer's capability score using expertise, performance, and availability (c) build metrics for relative bug importance score. Results of the experiment on five open-source projects (Mozilla, Eclipse, Net Beans, Jira, and Free desktop) demonstrate that with the proposed approach, roughly 20% of improvement can be achieved over existing approaches with the harmonic mean of precision, recall, f-measure, and accuracy of 92.05%, 89.04%, 90.05%, and 91.25%, respectively. The maximization of the throughput of the bug can be achieved effectively with the lowest cost when the number of developers or the number of bugs changes. The proposed solution addresses the following three goals: (i) improve triage accuracy for bug reports, (ii) differentiate between active and inactive developers, and (iii) identify the availability of developers according to their current workload.

Corresponding authors
Chetna Gupta,
chetnagupta04@gmail.com
Varun Gupta,
varun.iit13@gmail.com

## INTRODUCTION

The problem of assigning bugs is a challenging task aiming to identify potential developers based on bug meta-data features, the developer's performance profile, and other concerns such as workload schedule, technical aspects, and related documentation. Large software projects, including industrial and open-source projects, use bug tracking systems (BTS)

like Bugzilla (https://www.bugzilla.org/) to manage bug fixing. Effective bug assignment needs a plethora of information about reported bugs, their priority, severity, the pool of available developers, and information about their ability, experience, workload, availability, schedules, and bug report dependencies, among other things (*Zhang et al., 2017*; *Soltani, Hermans & Bäck, 2020*). Some of these parameters, such as severity and priority, are subjective and determined by the user's or developer's intuition rather than by a clear-cut framework. Many people who report software problems do not know the precise technical terms used in software development, which makes the bug unlabeled, vague, and noisy. It is considered a complex, multi-criteria decision-making process. Any mistake in this regard will cause an increase in overall bug-fixing time. In large open-source projects with a high number of daily bug reports, bug triagers find it challenging to keep track of all the developers and their progress. Thus, bug triagers face the dilemma of "How to select a potential developer from the pool of available developers for bug assignments under numerous constraints and achieve a timely and effective bug resolution solution?"

In addition to the traditional method of manually assigning bugs to appropriate developers, the literature provides an exhaustive array of approaches for semi- or fully automating the bug assignment process. These techniques are based on machine learning, social network analysis, tossing based on the graph, fuzzy logic, games in software development, operational research, and information retrieval, *etc.*, to perform automatic text summarization, duplicate detection, bug triaging, component prediction, severity/ priority prediction, *etc.*, (*Sajedi-Badashian & Stroulia, 2020*; *Nagwani & Suri, 2023*). These approaches try to minimize the effort, bug tossing length, and time required for bug resolution. To our knowledge, the literature lacks studies supporting multi-criteria decision-making for assigning a bug to the most potential developer, focusing on automated hybrid methods. The proposed approach considers (a) metrics for computing developer's capability (or expertise) score concerning relative performance in the past along with their availability status; (b) metrics for the relative bug score value; and (c) focusing on how to increase software developer's motivation. Decisive and explicit knowledge about the developer's performance profile and bug importance are computed from meta-features of the bug reports. It helps focus on the maintenance process to manage projects more effectively and efficiently.

The proposed approach uses intuitionistic fuzzy sets (IFS) (*Atanassov, 1986*) and a multi-objective evolutionary algorithm—particle swarm optimization (MPSO), to recommend appropriate developers. A multi-purpose evolutionary algorithm (EA) serves multiple optimization objectives at the same time, such as multi-objective optimization, solution exploration, noise robustness, complexity handling, and scalability. In bug assignment processes, it balances competing objectives such as bug severity and developer responsibilities while navigating noisy environments and tackling complex problems quickly. The algorithm scales well for large-scale activities, making it a useful tool for optimizing bug assignment processes and other comparable tasks. Multi-objective evolutionary algorithms (MOEAs) efficiently balance conflicting goals in bug assignment, analyze trade-offs, tackle complexity, navigate uncertainties, and scale well for large-scale

tasks. It is a collaborative approach wherein it builds metrics of developer's capability scores to provide ranking to available developers based on their performance, expertise, and availability from the pool of developers. It is a relative performance score value that helps analyze the relationship between the developer's capability concerning bug assignment tasks taken in the past. Next, a bug value score metric is computed using Intuitionistic fuzzy sets. From the available list of multi-criteria decision models, IFS is a powerful method that leverages the benefits of uncertain and vague human decisions clearly and intuitively, a scalar value that considers both the best and worst options, simple and easy to use with good computational efficiency. In the past, fuzzy logic-based approaches have not considered multi-criteria to compute bug value scores. Therefore, maximizing the developer's capability and bug scores to achieve a successful bug assignment is the objective. Next, it optimizes the two score lists (of developers and bugs) using an evolutionary algorithm (particle swarm optimization). Optimization helps find the best value of decision variables, which will be used to measure a decision's effectiveness. Overall, the proposed approach benefits the proposed approach handles two significant issues: differentiating between active and inactive developers and confusion over the assignment of bugs. It will further reduce bug-fixing delays and will prevent re-assignment problems.

The proposed method is applied and tested on five well-known open-source bug repositories. The results are compared to the state-of-the-art approaches to evaluate the best prediction accuracy and address the issue of reduced bug tossing length. The proposed work's performance is evaluated against the results of the fuzzy logic-based Bugzie model (*Tamrawi et al., 2011*). The proposed solution aims to reduce triagers' effort by addressing the increase in daily bug reports, particularly in large-scale open-source projects. With a rising number of bug reports flooding bug repositories, each bug must be triaged, resulting in lengthier repair times and a higher probability of reassignment. Automated bug triage provides a way to reduce the stress on triagers and speed up the procedure for fixing them (*Nagwani & Suri, 2023*). The following are the main contributions of this article:

- The use of fuzzy logic and an evolutionary algorithm is proposed as a new technique to improve the quality of the bug assignment.
- A metric is built to gather precise and explicit knowledge of the developer's capability score. Intuitionistic fuzzy logic is applied to multiple criteria to handle uncertainty and the vagueness of expert judgment to compute the bug value score of each bug. These two values will serve as input to the automatic process.
- With the proposed approach, maximization of the throughput of the bug and the assignment can be achieved concurrently by creating a balance between multiple selection and assignment criteria. It uses the evolutionary algorithm (particle swarm optimization) for selecting potential developers who can provide robust solutions with reduced overheads in cost, time of bug fixing, and bug tossing length.
- The proposed solution addresses the following three goals: (i) improve triage accuracy for bug reports, (ii) differentiate between active and inactive developers, and (iii) identify the availability of developers according to their current workload.

Specifically, the following research questions are investigated.

RQ1: How effective is a multi-objective particle swarm optimization (MPSO) based bug assignment technique?

(a) Can it improve the expert's manual assignments to minimize bug tossing length and timely resolution of bugs?
(b) How costly is it to compute developers' capability score, use IFS to compute bug value score, and run the multi-objective PSO-based approach in terms of accuracy?

RQ2: How valid and feasible are the solutions provided by a proposed approach, from:

(a) The human expert's perspective?
(b) What is the successful assignment rate?

RQ3: How helpful is the incentive mechanism in minimizing backlogs and overall predicted bug-fixing time?

The rest of the article is organized as follows: "Related Work" provides a discussion of existing literature, followed by a discussion of the proposed approach in "Proposed Approach". "Multi-Objective Particle Swarm Optimization for Solving Bug Assignment Problem" discussed the multi-objective particle swarm algorithm, followed by the results. "Empirical Validation" presents the conclusion and future work.

## RELATED WORK

The bug triaging problem has been the subject of several theories, some of which emphasize machine learning and information retrieval, auction-based, and social network approaches. In contrast, others emphasize fuzzy logic (*Soltani, Hermans & Bäck, 2020*; *Nagwani & Suri, 2023*). When it came to automating the software bug-triaging approach, machine learning (ML) techniques were the first to be considered by researchers. Machine learning approaches (*Tamrawi et al., 2011*; *Bhattacharya, Neamtiu & Shelton, 2012*; *Shokripour et al., 2015*; *Jonsson et al., 2016*; *Xia et al., 2017*; *Jiechieu & Tsopze, 2020*; *Mohsin & Shi, 2020*; *Tran et al., 2020*) match the new bug report with the characteristics closest to a set of bug reports fixed by a developer for a recommendation. From the literature listed, it can be concluded that most often, machine learning's classification technique (*Jiechieu & Tsopze, 2020*; *Mohsin & Shi, 2020*) is employed for bug triaging, and performance measures for classification tasks, including accuracy, precision, recall, and F1-measure, are used to assess the effectiveness of the proposed techniques. In a study of various machine learning techniques for detecting software flaws, *Tran et al. (2020)* discovered that the random forest classifier performs better than other techniques. Although machine learning (ML) is a widely used technology, gaining better performance is still a concern that encourages researchers to improve the current technologies (*Nagwani & Suri, 2023*). As reported by researchers, the accuracy achieved using machine learning algorithms ranges from 44.4% to 86.09%, with 86% being the highest level of precision.

Some previous research has attempted to match developer profiles' competence to a set of characteristic attributes to recommend developers who match their profile

(*Nagwani & Suri, 2023*; *Jahanshahi et al., 2021*; *Liu et al., 2022*)—the most complex challenge in such matching is labeling bug reports with insufficient or missing label information. Research has shown that using different aspects, such as the decision classifier, feature selection, tossing graphs, and incremental learning, impacts bug-triaging efficiency. Numerous machine learning techniques are employed in the literature to comprehend software bug reports and their attributes (*Kashiwa & Ohira, 2020*), along with their causes, such as dependencies between bugs (*Jahanshahi et al., 2021*; *Almhana, Kessentini & Mkaouer, 2021*; *Pan et al., 2022*; *Jahanshahi & Cevik, 2022*) and invalid and unreproducible bug reports (*Gundersen, Coakley & Kirkpatrick, 2022*; *Wu et al., 2020*).

Researchers have also explored techniques based on information retrieval to automate the bug-assigning process (*Tamrawi et al., 2011*; *Shokripour et al., 2015*; *Xia et al., 2017*; *Aung et al., 2022*). Bug reports are viewed as documents that are altered in these techniques. These methods use feature vectors to describe textual information in bug reports, which is then processed to determine which developers should be considered. The key idea is to assign a bug to a developer with comparative skill in dealing with a specific type of bug based on the developers' previous work. The most popular presentations in IR-based approaches are TF-IDF (term frequency—inverse document frequency), text mining, and text similarity techniques. *Guo et al. (2020)* used Word2vec, a natural language processing system, to summarize bugs and CNN to implement it. To increase the accuracy of bug assignments, a few researchers have employed extra information like components, products, severity, and priority (*Xia et al., 2017*; *Zhao et al., 2019*; *Yadav, Singh & Suri, 2019*).

Additionally, the researchers have suggested topic models for better software issue triaging. The topic model latent Dirichlet allocation (LDA) approach is used in *Zhao et al. (2019)* to identify a suitable developer. They use the LDA approach to calculate the similarity of bug reports and combine it with multiple attribute information to filter out inconsistencies. To obtain a closer supervised subject distribution, *Xia et al. (2017)* suggested a model adding additional supervision information to the LDA. However, there are fears that bug reports will be labeled with insufficient, missing, or redundant label information, resulting in the loss of context information. Another concern is the cost of using machine learning or information retrieval techniques.

There is limited work that addresses the developers involvement in the bug assignment process. It owes to the fact that there are so many developers that there is no way to know who is available, who has left the job, or who has the skill or potential to solve a given bug. In this regard, *Xuan et al. (2017)* suggested a semi-supervised text categorization algorithm to recommend that developers flag erroneous developer situations in current bug report data. Their method combines the naive Bayesian and expectation-maximization approaches. *Gupta, Inácio & Freire (2021)*, along with *Jahanshahi et al. (2021)*, *Jahanshahi & Cevik (2022)*, and *Gupta & Freire (2021)*, are among a few researchers who have proposed approaches taking into account the developer's involvement while allocating bugs. An auction-based blockchain framework is proposed by *Gupta & Freire (2021)*. Their work uses a blockchain-based incentive system for assigning bugs. Individual developers

can choose the bug reports that best suit their tastes and availability to provide reliable fixes with less expense and bug-fixing time.

*Xuan et al. (2012)* use social networking approaches to prioritize developers by analyzing developer information. To assign bugs depending on the developer's priority, they looked at three primary influencing factors: product characteristics, time fluctuation, and noise tolerance. A developer's social network depicts social interactions and personal relationships among software developers in social network-based methodologies. The developer's expertise is calculated based on the network's numerous influencing elements, which include software developer relationships and bug reports for the prospective developers (*Zhang et al., 2013*; *Alazzam et al., 2020*). The main issue with these strategies is that creating association graphs and aggregating data from various sources is difficult (*Alazzam et al., 2020*). Modeling social network analysis-based techniques is challenging since they use graph data structures for developer-bug relationships. Because of this, the computing time is longer.

Tossing graph-based techniques (*Bhattacharya, Neamtiu & Shelton, 2012*; *Chen, Wang & Liu, 2011*; *Jeong, Kim & Zimmermann, 2009*; *Bhattacharya & Neamtiu, 2010*) are another group of approaches mentioned in the literature. The tossing paths of previously repaired bug reports are considered in these approaches. To improve the accuracy of bug report assignments, *Jeong, Kim & Zimmermann (2009)* used the transfer graph to describe the bugs the current developer could not repair and the information that the bug report passed on to other developers. By examining transfer graphs mixed with various features, *Bhattacharya & Neamtiu (2010)* presented an enhanced assignment accuracy approach based on *Sajedi-Badashian & Stroulia (2020)*. One of the most severe issues in bug triaging is bug tossing, which accounts for around 93% of all bug reports tossed at least once.

The literature presents multiple solutions using mathematical and optimization approaches (*Wei et al., 2018*; *Kashiwa & Ohira, 2020*; *Kumar et al., 2020b*), including a few recent ones. The researchers primarily focus on creating mathematical formulations and objective functions to handle software bug triaging using mathematical modeling and optimization-based techniques. The advantages of these techniques include the potential for performance modeling and the ability to scale them by considering additional data and features when mathematically modeling bug-triaging processes. Most software bug features are textual, and fuzzy logic is the discipline closely resembling textual situations and is motivated by human intelligence. Fuzzy logic is the ideal method for determining the relationship between each developer and software issues because modern software development is based on a multi-developer, multi-tasking team. The fuzzy sets-based algorithm calculates the membership score of developers for specific topics based on bug parameters (*Xia et al., 2017*; *Chawla & Singh, 2015*). Using fuzzy logic and similarity measurements, these methods classify bug reports into bugs and non-bugs.

From the literature using fuzzy logic-based approaches, it can be concluded that by managing the various causes of software issues, fuzzy logic performs better than machine learning algorithms and offers a more significant number of developers to the triager for fixing newly reported bugs. Software defects are investigated and analyzed using fuzzy similarity measures (*Liu et al., 2020*; *Coletti & Bouchon-Meunier, 2019*), classified

(*Chen et al., 2019*; *Pandolfo et al., 2020*). In addition, models for triaging bugs based on fuzzy logic are created (*Tamrawi et al., 2011*; *Elbeltagi, Hegazy & Grierson, 2005*). *Tamrawi et al. (2011)* suggested a bug distribution technique based on fuzzy sets and developer caching. Additionally, previous research did not allocate problems based on critical criteria that considered the total developer's capability and knowledge. One of the main issues with distinguishing active and inactive developers is the lack of sufficient research on the availability of developers in the literature. This study fills in the gaps and suggests a hybrid bug-triaging approach.

## PROPOSED APPROACH

The proposed approach is an iterative method that can handle single and multiple human decision-makers (triagers) and different preference criteria. An idiosyncrasy of this approach is the use of IFS and evolutionary algorithms to reduce the manual effort, that is, the amount of information processing required from bug triagers to decide who a particular bug is to be allocated among the developers. A finite number of bugs $B = \{b_1, b_2.....b_n\}$ and a finite number of developers $D = \{d_1, d_2.....d_n\}$ is considered for bug assignment. The foremost requirement of the presented approach is to have a simple and fast process of analysis yielding accurate and trustworthy results. If both of these conditions are not met, the process is unlikely to be used in the bug-triaging process. Literature has established that of all the criteria, three main factors are the quality of bug resolution, the developer taking the average time to fix the bug, and the priority of bug report. For a successful bug resolution, the quality of bug resolution must be maximized, time-to-fix must be minimized, and the priority of bug reports should be maximized. Figure 1 sketches the proposed bug assignment process.

As shown in Fig. 1, the features of both developer profiles and bug reports are studied—the benefit of analyzing and addressing bug reassignment issues. Later, score values for both are calculated using the proposed capability score and bug value score (discussed next in this section). These scores are then sorted to form the priority list, which is inputted into the algorithm to optimize results.

In this article, a multi-objective particle swarm optimization (MPSO) algorithm is used because it has a better success rate, quality of the solution, and processing time in comparison to other evolutionary algorithms such as ant-colony systems, memetic algorithms, genetic algorithms, and shuffled frog leaping (*Wei et al., 2018*). The initial investigation focused on five key databases: Mozilla, Eclipse, NetBeans, Jira, and Free Desktop. It was found that bug reports across these databases share common labels and attributes, including bug description, comments, attachments, dependencies, bug ID, creation date, reporter name, product, components, priority, platform, assignee name, bug history, bug status, keywords, version, operating system, and severity.

As discussed in related work, the existing methods do not explicitly consider multiple features and the amount of workload a developer can handle in a given time. It may result in work overload for a few developers. However, these techniques have recommended appropriate developers or produced effective results in shortening bug fixing or tossing length. To tackle this problem, in this article, developers' capability, bug importance, and

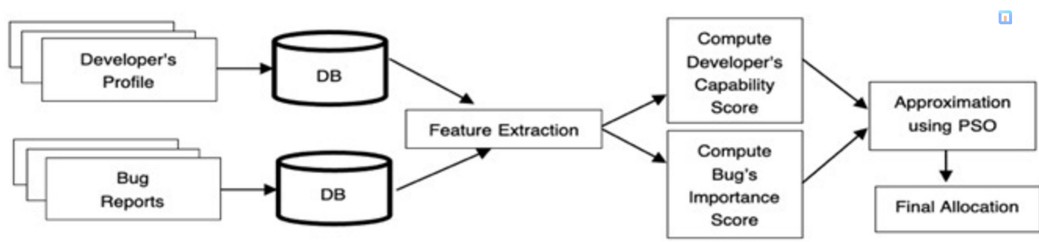

**Figure 1 Bug assignment process using the proposed method.**

availability of a developer are considered using a multi-criteria approach. The results are optimized to recommend the appropriate developer for the reported bug. Based on the count of bug fixes relating to the component indicated in the reported bug, the proposed method measures bug-fixing competence and performance in addition to their current workload. The following steps are followed:

### Developer's capability score

An in-depth analysis of developers' performance and expertise in resolving the bug is considered. Developers are ranked according to their relative performance in handling bug resolution in the past. A list of features is extracted for each developer as a function of their profile expertise regarding bugs handled. The capability score $(D_{cap})$ is defined as the percentage of successfully resolved bug reports $(S_{R(bi)})$ with the total number of bug reports $(T_{(bi)})$ handled, mathematically formulated in Eq. (1). Total bug reports include both successful and unsuccessful reports handled.

$$D_{cap} = \left( SR_{(bi)} / T_{(bi)} \right) \times 100 \tag{1}$$

In short, the proposed approach provides the solution to the problem of automated bug triaging using the following essential philosophy: "*Who has the most bug-fixing capability score (relative score) for (i) current workload (ii) the relative performance of handling bug fixing record in the past.*" Developers will be ranked based on their capability score values from a filtered list. This ensures that a developer can always be selected or recommended with a possibility of 1, even in the worst-case scenario where there is no suitable developer available. Algorithm I discuss the steps followed to build the capability score of available developers.

In step 4 (referring to Algorithm I), the upper and lower threshold values are computed: A developer can handle a limited number of bugs in a day or month. The value of the *current load* (total bug-fixing cost assigned) refers to the load a developer is handling. This parameter is used as a parameter to set the upper threshold. The lower threshold is the minimum amount of workload a developer is handling. It also includes inactive developers. The threshold values for both upper and lower are already set in advance. In experimentation, the same threshold values for all developers are considered, but in practice, these can be altered depending on the number of bugs or the project.

| Algorithm I: | Compute capability algorithm |
|---|---|
| 1. | Compile a list of developers and the number of bugs allocated in the past. |
| 2. | Generate a separate list of bug-fixing data. |
| 3. | Filter the results for each developer using the attributes listed below: |
| a. | Fetch: Get the fixer information and bug description from the data (title, keywords, product name, and component). |
| b. | *Maxalloc*: the maximum number of bugs assigned so far. It consists of the following: |
| |    i. *Bugs assigned*<br>   ii. *Bugs resolved/tossed* |
| c. | *Current load*: the number of currently assigned bug reports. It also represents the status of a developer as active or inactive. |
| 4. | Filter developers according to the capacity to take the new load (developers are isolated using an upper and lower threshold value). |
| 5. | Compute capability score using Eq. (1). |

| | |
|---|---|
| $C(b_i)$ | It is the complexity of the bug. |
| $V(b_i)$ | It represents its volatility, the approximate time over which it cannot be changed. |
| $Tm(b_i)$ | Estimated time/duration for resolving a specific bug. It is influenced by factors such as the environment, support for infrastructure, *etc.* |
| $S(b_i)$ | Is the degree of the impact a bug can have on the system |
| $P(b_i)$ | Is the priority of fixing the bug |
| $E(b_i)$ | Estimated effort required to resolve bugs within constraints of the environment and technical skill required. |
| $Tn(b_i)$ | When a bug/feature is not resolved or missed within a specified time and duration. It is the number of times the bug is tossed to another developer. |

## Bug value computation

Bug value computation refers to the process of assessing the importance and impact of bugs in software development. It involves analyzing various parameters such as bug complexity, volatility, time to fix, severity, priority, effort, and tossing information to determine the value or significance of each bug in the bug resolution process. This computation helps bug triager's prioritize bug resolution effectively and allocate resources accordingly. A multi-objective model is considered on the following features extracted from the available bug repository: bug complexity, volatility, time to fix, severity, priority, effort, and associated tossing information. Based on the definition of various parameters of standard bug reports, Let B be a set containing n bugs, and for each bug $b_i \in B$ let,

For effective bug resolution, it is essential to perform bug report estimations and access their potential payoffs before allocating them to developers. No two bug reports are the same; each is unique in what it sets out to achieve and unique in the multitude of parameters that form its existence. Usually, bug triagers face difficulty in deciding or identifying which bug is to be assigned to which developers. Frequently, a problem that seems simple initially turns out to be more complex or technically challenging to solve.

Different perspectives and approaches to reporting bugs and assigning them can lead to varied outlooks. Therefore, predictions cannot rely solely on linguistic values assigned to bug report labels. Hence, using multi-criteria decision analysis, the Intuitionistic Fuzzy Logic (*Atanassov, 1986*) measure is used to interleave human actions. The primary reason for using intuitionistic fuzzy logic over linguistic values is that it can handle uncertainty and vagueness. It is significantly closer to how people (in this case, bug triager) put across and use their insight to rank any item (here, bug). In the context of intuitionistic fuzzy logic systems (IFS), bug value computation is utilized as part of a multi-criteria decision analysis framework. Bug value computation within an IFS setting involves assigning linguistic variables to the various parameters of bugs (such as complexity, volatility, severity, *etc.,*) and then applying fuzzy logic techniques to analyze these variables.

If we look closely at linguistic values, IFS is an extension of linguistic values, describing linguistic variables in a detailed manner. Hence, it is rational and practical to use IFS to predict risk to avoid uncertainly grasping these values by the decision-makers. The values for these criteria are fetched from bug reports. A brief overview of the functionality of IFS (*Atanassov, 1986*) is discussed below:

In IFS, the inputs are in the form of membership and non-membership and are defined as:

$$F = \{x, \mu_F(x), \upsilon_F(x) | x \in F\}$$

with a degree of membership and non-membership for the element $x$ as

$$\mu_F : X : [0, 1]$$
$$x \in X \rightarrow \mu_F(x) \in [0, 1]$$

and

$$\upsilon_F : X : [0, 1]$$
$$x \in X \rightarrow \upsilon F(x) \in [0, 1]$$

such that all values of $x$ in X will be confirmed in the following equations

$$0 \leq \mu_{F(x)} + \upsilon_F(x) \leq 1 \tag{3}$$

hesitation index is defined as,

$$\pi_F(x) = 1 - \mu_F(x) - \upsilon_F(x) \tag{4}$$

such that for every $x \in X$

$$0 \leq \pi_F(x) \leq 1 \tag{5}$$

The degree of membership of any element, $x$, can also be re-formulated in the closed interval range as:

$$\left[\mu_F^l, \ \mu_F^u\right] = \left[\mu_F, \ \mu_F + \pi_F\right] \tag{6}$$

the evaluation of the alternative $x_j \in X$ with respect to the attribute $a_i \in A$ is an intuitionistic fuzzy set, where $X_{ij} = \{<x_j, \mu_{ij}, \upsilon_{ij}\}$. In the intuitionistic indices $\pi_{ij} = 1 - \mu_{ij}$,

$\upsilon_{ij}$ the larger $\pi_{ij}$ represents a higher hesitation margin of the decision maker as to the "excellence" of the alternative $x_j \in X$ concerning the attribute $a_i \in A$ whose intensity is given by $\mu_{ij}$. These intuitionistic indices are significant as they are used to calculate the best final and the worst-case result. Using Eq. (6), decision-makers can adjust their evaluation by adding the value of the intuitionistic index. As shown in Eq. (6), the evaluation lies in a closed interval, where $\mu_{ij}^l = \mu_{ij}$ and, $\mu_{ij}^u = \mu_{ij} + \pi_{ij} = 1 - \upsilon_{ij}$. Also, $0 \le \mu_{ij}^l \le \mu_{ij}^u \le 1$ for all $x_j \in X$ and $a_i \in A$. To attain values from bug reporters, for $m$ criteria, intuitionistic fuzzy set $B_{ij}$ can be represented as:

$$B_{ij} = \{r_i, c_j, \mu_{ij}, \upsilon_{ij}\} \tag{7}$$

where $0 \le \mu_{ij} \le 1, 0 \le \upsilon_{ij} \le 1$ and $0 \le \mu_{ij} + \upsilon_{ij} \le 1$

also, $0 < i \le n$ and $0 < j \le m$

where $n$ = total number of bugs in the given set; $m$ = total number of criteria, $\mu_{ij}$ & $\upsilon_{ij}$ represents the degree of membership and non-membership for the bugs $b_i \in B$ for the criteria $c_j \in C$. The higher hesitation index value represents the higher hesitation of the bug reporter to decide upon $b_i \in B$ for the criteria $c_j \in C$.

$$\pi_{ij} = 1 - \mu_{ij} - \upsilon_{ij} \tag{8}$$

where $0 < i \le n$ and $0 < j \le m$

Given intuitionistic fuzzy values, each criterion is assigned a weight. For each alternative bug $b_i \in B$, the optimal rank value can be computed using the following equation:

$$\max\left\{ p_i = \sum_{j=1}^{m} \alpha_{ij}\omega_j \right\} \tag{9}$$

such that: $i = (1,2,3\ldots\ldots n)$

$$\mu_{ij}^l \le \alpha_{ij} \le \mu_{ij}^u \tag{10}$$

$$\omega_j^l \le \omega_j \le \omega_j^u \tag{11}$$

$$\sum_{j=1}^{m} \omega_j = 1$$

To compute optimal rank value using Eq. (9), two linear programming equations can be derived:

$$\min\left\{ p_i^l = \sum_{j=1}^{m} \mu_{ij}^l \omega_j \right\} \tag{13}$$

such that:

$$\left\{ \begin{array}{c} \omega_j^l \le \omega_j \le \omega_j^u \\ \sum_{j=1}^{m} \omega_j = 1 \end{array} \right\} \tag{14}$$

and

$$max\left\{p_i^u = \sum_{j=1}^{m} \mu_{ij}^u \omega_j\right\}$$

such that:

$$\left\{\begin{array}{l} \omega_j^l \le \omega_j \le \omega_j^u \\ \sum_{j=1}^{m} \omega_j = 1 \end{array}\right\}$$

for each $i = (1, 2, 3...n)$

Solving the above two linear programming equations using the Simplex Method, the following optimal solution of the criteria weights can be computed.

$$\omega^{i'} = \left(\omega_1^{i'};\ \omega_2^{i'}; \omega_3^{i'} \ldots\ldots\ldots \omega_m^{i'}\right)$$

and

$$\omega^{i''} = \left(\omega_1^{i''}; \omega_2^{i''}; \omega_3^{i''} \ldots\ldots\ldots \omega_m^{i''}\right)$$

With the help of these optimal values of weights for criteria, the optimal solutions for the rank value of the bug can be found as follows:

$$p_i^{l'} = \sum_{j=1}^{m} \mu_i^l \omega_j^{i'} = \sum_{j=1}^{m} \mu_{ij} \omega_j^{i'} \tag{15}$$

and

$$p_i^{u''} = \sum_{j=1}^{m} \mu_i^u \omega_j^{i''} = 1 - \sum_{j=1}^{m} \nu_{ij} \omega_j^{i''} \tag{16}$$

for each $i = (1, 2, 3...n)$

The comprehensive values of all the alternatives according to the equations listed above will be different because of the different values of the weight vectors. Hence, all the bugs in '$B$' are non-inferior to each other, and the objective function can be $p_i^l$ can be re-written for every bug $b_i \in B$ as

$$min\left\{p_t^l = \frac{\sum_{i=1}^{n}\sum_{j=1}^{m} \mu_{ij}^l \omega_j}{n}\right\} \tag{17}$$

also,

$$max\left\{p_t^l = \frac{\sum_{i=1}^{n}\sum_{j=1}^{m} \mu_{ij}^u \omega_j}{n}\right\} \tag{18}$$

$$\left\{ \begin{array}{c} \omega_j^l \leq \omega_j \leq \omega_j^u \\ \sum\limits_{j=1}^{m} \omega_j = 1 \end{array} \right\}$$

Using these weight vectors, the final optimal value for all the alternatives can be obtained with the following formula:

$$\beta_i = \frac{\sum\limits_{j=1}^{m} \mu_{ij}^u \omega_j'}{1 + \sum\limits_{j=1}^{m} \left( \mu_{ij}^u - \mu_{ij}^l \right) \omega_j'} \tag{19}$$

Equation (19) is rewritten using Eqs. (15) and (16), defining the lower and upper intervals of the priority value of the bug. Here, $\beta_i$ represents the final optimal value of the bug, which can be calculated by using the distance between the intervals of the membership value of the bug concerning every criterion $\mu_{ij}^u$ and the weight of the criteria. Using this method, bug triggers provided values for the degree of membership and non-membership for every bug according to the number of criteria considered.

The degree of membership and non-membership can lie anywhere within the interval [0, 1]. The 0 value represents the value that does not belong to the given fuzzy set, and 1 represents the value that entirely belongs within the fuzzy set. Any value between 0 and 1 represents the degree of uncertainty that the value belongs in the set. In short, the following is the crucial outcome: "*Which is the most important bug of all the reported bugs which requires urgent attention?*"

## Illustrative example

Consider an illustrative example of having ten bugs to be considered for assignment. Using intuitionistic inputs, the hesitation index and optimal weights for each criterion are computed using linear programming presented in Table 1. The values of membership (M) and non-membership (NM) used are as follows:

{C1:M = 0.25; NM = 0.6}, {C2:M = 0.1; NM = 0.65}, {C3:M = 0.35; NM = 0.5}, {C4:M = 0.2; NM = 0.5}, {C5:M = 0.35; NM = 0.5}, {C6:M = 0.7; NM = 0.1}, {C7:M = 0.3; NM = 0.5}.

Referring to Table 1, decision-makers (here, bug triage) provide inputs for computing the bug priority values based on the selected criteria for each bug. The bug triagers will specify the membership degree and non-membership degree values for a bug for each criterion in the range of 0–1.

In case of more than one bug triager or decision-maker, aggregated score value of weights will be considered. Here, C1–C7 represents the criteria as C(bi), V(bi), Tm(bi), S(bi), P(bi), E(bi), and Tn(bi), respectively. These intuitionistic inputs help in computing the hesitation index and optimal weights of each criterion through linear programming. The criteria weights and inputs provided by the bug triager are reported in Table 1. These are used to calculate the optimum ranks of each bug according to Eq. (18) obtained above. The weights of these criteria are computed using the Simplex method (refer to Table 2) to compute the optimal rank of the bugs.

**Table 1  Bug membership and non-membership values.**

| Bugs | C1 | | C2 | | C3 | | C4 | | C5 | | C6 | | C7 | |
|------|------|------|------|------|------|------|------|------|------|------|------|------|------|------|
| | M | NM | M | NM | M | NM | M | NM | M | NM | M | NM | M | NM |
| B1 | 0.5 | 0.3 | 0.55 | 0.25 | 0.6 | 0.3 | 0.15 | 0.65 | 0.15 | 0.65 | 0.45 | 0.35 | 0.65 | 0.15 |
| B2 | 0.45 | 0.1 | 0.5 | 0.05 | 0.45 | 0.5 | 0.65 | 0.15 | 0.65 | 0.15 | 0.5 | 0.4 | 0.3 | 0.4 |
| B3 | 0.25 | 0.35 | 0.2 | 0.55 | 0.5 | 0.4 | 0.35 | 0.5 | 0.35 | 0.5 | 0.5 | 0.4 | 0.5 | 0.4 |
| B4 | 0.1 | 0.75 | 0.55 | 0.25 | 0.35 | 0.55 | 0.5 | 0.4 | 0.5 | 0.4 | 0.35 | 0.4 | 0.3 | 0.4 |
| B5 | 0.05 | 0.75 | 0.35 | 0.25 | 0.5 | 0.3 | 0.65 | 0.25 | 0.65 | 0.25 | 0.5 | 0.25 | 0.5 | 0.25 |
| B6 | 0.6 | 0.1 | 0.35 | 0.3 | 0.65 | 0.2 | 0.5 | 0.45 | 0.5 | 0.45 | 0.5 | 0.35 | 0.25 | 0.35 |
| B7 | 0.3 | 0.5 | 0.25 | 0.55 | 0.1 | 0.7 | 0.6 | 0.2 | 0.6 | 0.2 | 0.3 | 0.6 | 0.3 | 0.6 |
| B8 | 0.4 | 0.5 | 0.75 | 0.1 | 0.8 | 0.05 | 0.8 | 0.1 | 0.8 | 0.1 | 0.5 | 0.4 | 0.35 | 0.4 |
| B9 | 0.4 | 0.45 | 0.25 | 0.6 | 0.1 | 0.75 | 0.6 | 0.25 | 0.5 | 0.2 | 0.1 | 0.7 | 0.2 | 0.5 |
| B10 | 0.7 | 0.2 | 0.75 | 0.1 | 0.8 | 0.05 | 0.8 | 0.1 | 0.85 | 0.1 | 0.85 | 0.1 | 0.75 | 0.15 |

**Note:**
B1 to B10, Example 10 Bugs; C1–C7, Criteria 1 to 7; M, Membership values; NM, Non membership values.

**Table 2  Weights of criterion solved using the simplex method.**

| N | M | c1 | c2 | c3 | c4 | c5 | c6 | c7 |
|------|------|------|------|------|------|------|------|------|
| 10 | 7 | 0.2 | 0.15 | 0.12 | 0.12 | 0.15 | 0.16 | 0.1 |

**Note:**
N, Number of Bugs; C1, Complexity; C2, Volatility; C3, Estimated Time; C4, Degree of impact; C5, Priority; C6, Estimated Effort; C7, Tossed.

As discussed in "Introduction", these criteria are company/project dependent and may vary per project specification. The weights have been computed using values of membership (M) and non-membership (NM) mentioned above and using Eqs. (13) and (14). These weights are finally used to compute $p^u$ and $p^l$ according to Eqs. (15) and (16). Table 3 presents the results of the final rank values for each bug.

## MULTI-OBJECTIVE PARTICLE SWARM OPTIMIZATION FOR SOLVING BUG ASSIGNMENT PROBLEM

Given the bug value rank and developers' capability score rank, a particle swarm optimization algorithm is used to solve the bug assignment problem in an automated manner. The multi-objective PSO (MPSO) algorithm tries to find the best solution for defined multi-objectives by progressing towards optimizing a problem iteratively by improving the candidate solution.

Traditional optimization approaches require different problem functions and must be repeated several times to identify solutions. In recent years, the multi-objective optimization evolutionary algorithm has attracted scholarly interest as a popular method for solving multi-objective optimization problems involving several conflicting objectives. The particle's position is defined as a single solution, and its position is changed according to its own experience and that of its neighbors. The main objective is to direct the particle towards the most favorable fitness value of the particular solution/particle. The position of the $p_s(i)$ is changed by adding a velocity $v(i)$ to it as mentioned in the following equations:

**Table 3 Bug triager's bug priority values.**

| Bugs | $p^u$ | $p^l$ | $p^u$-$p^l$ | $\beta^a$ | Normalized value |
|------|-------|-------|-------------|-----------|------------------|
| B1 | 0.38 | 0.43 | 0.05 | 1.05 | 0.36 |
| B2 | 0.23 | 0.50 | 0.27 | 1.27 | 0.18 |
| B3 | 0.44 | 0.36 | 0.08 | 1.08 | 0.41 |
| B4 | 0.47 | 0.37 | 0.10 | 1.10 | 0.42 |
| B5 | 0.36 | 0.43 | 0.07 | 1.07 | 0.33 |
| B6 | 0.30 | 0.49 | 0.19 | 1.19 | 0.25 |
| B7 | 0.48 | 0.35 | 0.13 | 1.13 | 0.42 |
| B8 | 0.25 | 0.62 | 0.37 | 1.37 | 0.18 |
| B9 | 0.49 | 0.31 | 0.18 | 1.18 | 0.42 |
| B10 | 0.12 | 0.78 | 0.66 | 1.66 | 0.07 |

Note:
$\beta$ = Represents the final optimal value of the bug.

$$v(i+1) = w * v(i) + c_1 * rand(\ ) * (pbest - ps(i)) + c2 * rand(\ ) * (gbest - ps(i)) \quad (20)$$
$$ps(i+1) = present(i) + v(i) \quad (21)$$

The position vector $p_s$ represent the current particle (solution). Velocity $v$ is the particle velocity and is defined as the factor by which the priority value of the bug will change. *pbest* and *gbest* are those priority vectors considered as the optimal or best solution of the iteration. *rand ()* is a random number ranging from 0–1 such that $rand() \in [0,1]$, *c1* is a *cognitive parameter*, which denotes the particle's last position visited, *c2* is a *social parameter*, which highlights information gathered about the neighboring best position from social interaction. *w* is the inertia weight that controls the convergence behavior of PSO. For calculating approximate position values of all particle positions, $p_s = (p_1,p_2,p_3...p_n)$, Eqs. (20) and (21) are used. All particle continuous position values are converted to discrete vectors $dis(p_s) = (d_1, d_2, d_3..d_n)$ by applying the smallest position value (*Gupta & Freire, 2021*). Every particle here has *n-dimensional* space, $n$ bugs for allocation to $m$ developers, and has the following fitness functions: (a) maximizing capability score, (b) maximizing bug value score. Every particle will be assessed considering these fitness functions and all Pareto optimal solutions stored in a log using Eq. (22).

$$FA(present) = \sum_{n=1}^{m}(wt_n f_n(present)), \{\forall\ (p_s) \in Log\} \quad (22)$$

where FA is the final analysis, $m$ is the number of objective functions and $wt_n$ is the preference weight for every objective function $f_n(present)$. Pareto optimal solutions are ranked (log members) based on the number of functions that they minimize and maximize. Then $g_{best(n)}$ is randomly chosen from the top ten. Table 4 represents one of the possible solutions where developers are assigned bugs after converting position values to discrete values. It can be seen that $d_7, d_9, d_{10}, d_1, d_4, d_8....d_m$ are chosen to be assigned to bugs $b_1...b_n$ respectively. The particle position represents developers to whom specific bugs will be assigned, and the number represents the developer number.

**Table 4 Final bug assignment after $i^{th}$ iteration.**

| Bugs | $b_1$ | $b_2$ | $b_3$ | $b_4$ | $b_5$ | $b_6$ | $b_7$ | $b_8$ | $b_9$ | $b_{10}$ |
|---|---|---|---|---|---|---|---|---|---|---|
| Developers | 7 | 9 | 10 | 1 | 4 | 8 | 5 | 3 | 2 | 6 |

## Empirical validation

In this section, we aim to prove the efficiency of our automated multi-objective bug assignment method. The evaluation consists of two parts: (1) an experimental study conducted on five significant databases, namely, Mozilla, Eclipse, NetBeans, Jira, and Freedesktop, and (2) a user study in which interviews with experts (bug triagers) were conducted to analyze their feedbacks and assess the feasibility of the approach in terms of simplicity and processing speed; yielding accurate and trustworthy results. Acquiring the ground truth ranking involves analyzing historical data from bug reports and developer performance metrics. Successful matches entail effective bug resolutions meeting requirements, while unsuccessful matches result in unresolved bugs. Continuous evaluation ensures accuracy and relevance.

## Data collection and experimental setup

Features such as bug ID, summary, bug reporter, comments, components, priority, severity time stamp, *etc.*, were extracted from bug reports. The bug reports from the following bug repositories, namely, Mozilla, Eclipse, Jira, Freedesktop, and NetBeans are considered along with their resolution status (labels) as RESOLVED, FIXED, CLOSED, and VERIFIED. The data is extracted from January 1, 2011, until November 30, 2022. The total developers count, keywords, product names, and component names were also extracted from each bug repository. All of these are well-known and well-established bug repositories. Existing studies have used these datasets to analyze bug reports. Because of this, this article uses the same dataset to validate the results obtained. Table 5 presents the statistics of the final filtered data. The filtered data consists of (a) bugs with known fixing times and (b) bugs having no outlier value as a fixing time. The collected data is pre-processed, and all stop words are removed.

Further, the noise from the dataset is reduced, and model execution is increased by eliminating (a) developers with less than 10 fixes (*Xia et al., 2017*), (b) frequently appearing words (more than 50% that appeared), and (c) too infrequently (less than ten times). The following procedure is followed for recommendation:

- *Step 1: Parameter setting*

  - Initialize the *upper* and *lower* thresholds and the *current load* value for each filtered developer.
  - Obtain the developers' capability and value scores every time a new bug is reported. The relevant features will be extracted, and the filtered results will be updated to compute both the score values. It is entirely an automated process. For providing

**Table 5 Statistics of filtered data.**

| Projects | Filtered bugs | Developers | Components | Product | Reporters |
|---|---|---|---|---|---|
| Mozilla | 9,581 | 145 | 375 | 21 | 1,509 |
| Eclipse | 8,390 | 70 | 32 | 10 | 1,100 |
| Jira | 9,882 | 75 | 45 | 9 | 1,147 |
| Freedesktop | 9,641 | 80 | 245 | 37 | 3,500 |
| NetBeans | 9,820 | 65 | 260 | 40 | 1,723 |

bugs, membership, and non-membership value ranges are already fixed in advance to assign these values.

- *Step 2: Optimization*

  ○ The multi-objective PSO will use two inputs, the bug value rank, and the developers' capability score rank, to build the best solution by maximizing the capability score and bug value score and employing Eq. (22).
  ○ On successful bug assignment, the value of the objective variable $(x_{ij})$ will be set to 1 otherwise 0 such that $x_{ij} \in \{0, 1\}$.

- *Step 3: Updating developers and bug list*

  ○ Update *current load* and *Maxalloc*
  ○ Bug data with status as *assigned*.

- *Step 4: Perform constraints check:*

  ○ Recommendation of only a few experienced developers matching expertise: This is tackled by setting a flag whenever the count of assignments for a developer (*i.e.*, *current load* value) reaches the upper threshold value.
  ○ Multiple recommendations to only a few developers: This is tackled by ensuring that at least one developer is recommended such that $x_{ij} \leq 1$

- *Step 5: Repeat all steps for the next allocation*

To simulate a real-life scenario, 80% of the data is used as the training set and the remaining 20% as the test set. The accuracy rate is used as the evaluation index to analyze the successful computation and ranking of developers and bugs @k accuracy, where k = Top 1 to Top 10. Since the analysis involves several criteria, this article uses several similarity measures to evaluate the performance of the proposed approach. An exhaustive comparison of performance is conducted against several benchmark classifiers, namely, Naive Bayes (NB), support vector machine (SVM), C4.5, and random tree (RT) under the same datasets. These classifiers are implemented by using Weka. The results are presented in Table 6.

**Table 6 Top-1 to Top-10 accuracies of various classifiers.**

| Bug repository | Classifier | Top-1% | Top-2% | Top-3% | Top-4% | Top-5% | Top-6% | Top-7% | Top-8% | Top-9% | Top-10% |
|---|---|---|---|---|---|---|---|---|---|---|---|
| Mozilla | NB | 25.78 | 30.19 | 35.45 | 36.79 | 42.71 | 47.08 | 50.63 | 56.15 | 59.80 | 61.48 |
| | SVM | 27.02 | 32.69 | 45.07 | 55.97 | 57.23 | 63.49 | 69.40 | 70.69 | 72.35 | 75.57 |
| | C4.5 | 26.40 | 32.47 | 37.86 | 52.43 | 55.91 | 57.32 | 58.26 | 64.00 | 65.94 | 70.03 |
| | RT | 25.81 | 35.31 | 36.43 | 39.51 | 39.55 | 43.37 | 43.54 | 46.37 | 66.72 | 69.17 |
| Eclipse | NB | 29.14 | 31.32 | 34.00 | 37.52 | 40.61 | 48.69 | 54.11 | 58.59 | 63.60 | 68.23 |
| | SVM | 32.12 | 37.54 | 44.34 | 50.38 | 57.47 | 66.42 | 69.48 | 72.13 | 77.52 | 78.19 |
| | C4.5 | 30.37 | 37.20 | 47.76 | 55.89 | 60.12 | 67.81 | 69.89 | 72.87 | 74.92 | 74.23 |
| | RT | 25.98 | 27.60 | 33.92 | 40.49 | 40.93 | 48.48 | 62.80 | 64.04 | 69.42 | 70.15 |
| Jira | NB | 26.25 | 29.15 | 37.57 | 41.47 | 48.09 | 58.38 | 63.70 | 64.76 | 65.15 | 66.47 |
| | SVM | 28.99 | 36.99 | 38.80 | 39.30 | 44.04 | 59.51 | 69.92 | 76.08 | 81.68 | 82.69 |
| | C4.5 | 29.70 | 39.20 | 42.23 | 44.26 | 44.45 | 52.68 | 61.39 | 62.23 | 70.40 | 71.95 |
| | RT | 28.92 | 62.60 | 78.47 | 41.08 | 57.20 | 43.90 | 44.17 | 22.50 | 72.75 | 66.11 |
| Freedesktop | NB | 27.41 | 32.16 | 40.72 | 43.98 | 44.70 | 59.67 | 62.39 | 63.83 | 67.21 | 69.17 |
| | SVM | 26.54 | 26.76 | 28.15 | 39.17 | 55.53 | 56.58 | 78.78 | 79.34 | 81.59 | 83.60 |
| | C4.5 | 32.84 | 38.72 | 41.80 | 45.63 | 48.36 | 53.99 | 56.20 | 59.93 | 61.23 | 67.61 |
| | RT | 25.02 | 30.05 | 41.40 | 43.75 | 49.56 | 50.14 | 53.51 | 59.59 | 60.83 | 69.32 |
| NetBeans | NB | 29.31 | 33.11 | 33.15 | 37.25 | 49.53 | 56.32 | 57.92 | 63.00 | 67.13 | 69.23 |
| | SVM | 29.32 | 35.50 | 41.33 | 43.53 | 50.07 | 57.94 | 64.21 | 69.90 | 73.12 | 79.91 |
| | C4.5 | 28.09 | 30.29 | 43.05 | 51.27 | 55.82 | 56.42 | 60.20 | 61.93 | 65.79 | 67.28 |
| | RT | 25.96 | 30.70 | 35.77 | 49.19 | 52.04 | 56.10 | 57.17 | 61.06 | 62.83 | 68.74 |

## Performance measure

The performance measures utilized are as follows:

(a) Bug assignment rate: This metric assesses the successful assignment of bugs to potential developers. It's computed as $B_g/B_t$, where $B_g$ represents the successfully assigned bugs, and $B_t$ is the total number of bugs to be assigned. A higher assignment rate indicates greater algorithm effectiveness.

(b) Accuracy: This measure represents the ratio of correctly predicted observations to the total observations.

(c) Precision: Referring to the fraction of relevant instances among the retrieved instances.

(d) Recall: This measures the fraction of retrieved relevant instances among all relevant instances. Precision and recall collectively gauge the measure of relevance.

(e) F-measure: This metric signifies the test accuracy and is derived from precision and recall values. It's calculated as: f-measure = (2 * Precision * Recall)/(Precision + Recall).

## RESULTS AND DISCUSSION

This section presents the results and discusses the computed results for all performance measures discussed in section VB. Firstly, the result analysis is the proposed ranking function is presented.

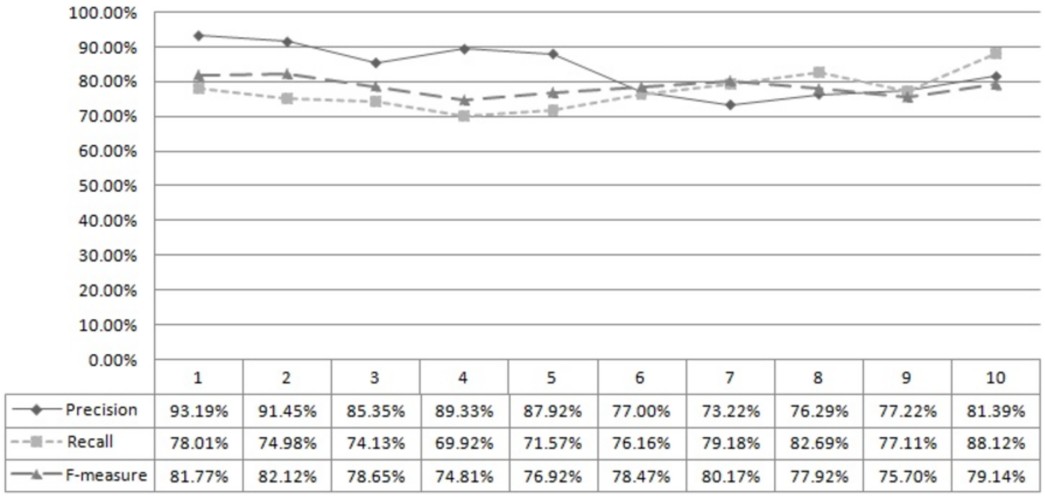

| | 1 | 2 | 3 | 4 | 5 | 6 | 7 | 8 | 9 | 10 |
|---|---|---|---|---|---|---|---|---|---|---|
| Precision | 93.19% | 91.45% | 85.35% | 89.33% | 87.92% | 77.00% | 73.22% | 76.29% | 77.22% | 81.39% |
| Recall | 78.01% | 74.98% | 74.13% | 69.92% | 71.57% | 76.16% | 79.18% | 82.69% | 77.11% | 88.12% |
| F-measure | 81.77% | 82.12% | 78.65% | 74.81% | 76.92% | 78.47% | 80.17% | 77.92% | 75.70% | 79.14% |

**Figure 2 Developer rank accuracy.**

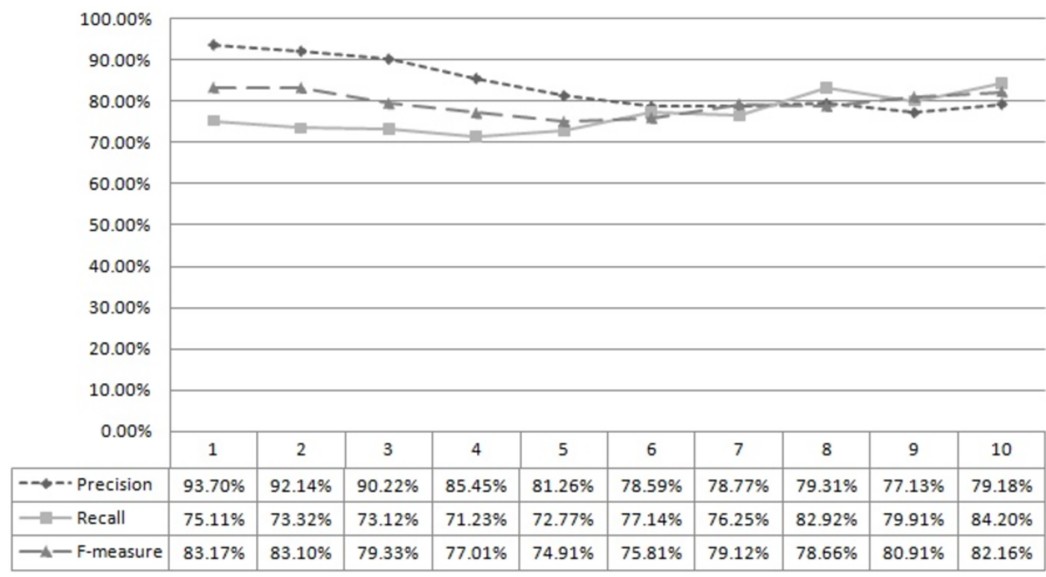

| | 1 | 2 | 3 | 4 | 5 | 6 | 7 | 8 | 9 | 10 |
|---|---|---|---|---|---|---|---|---|---|---|
| Precision | 93.70% | 92.14% | 90.22% | 85.45% | 81.26% | 78.59% | 78.77% | 79.31% | 77.13% | 79.18% |
| Recall | 75.11% | 73.32% | 73.12% | 71.23% | 72.77% | 77.14% | 76.25% | 82.92% | 79.91% | 84.20% |
| F-measure | 83.17% | 83.10% | 79.33% | 77.01% | 74.91% | 75.81% | 79.12% | 78.66% | 80.91% | 82.16% |

**Figure 3 Bug rank accuracy.**

## The overall accuracy of ranks generated by the proposed approach (RQ1)

Figure 2 shows developers' successful computation and ranking using the developer's capability score. A total of 400 test data sets were used to compute precision, recall, and f-measure. The first column, "x," describes the rank matching "x." For example, '5' means rank matching top 1 to 5. Similarly, Fig. 3 shows the successful computation and ranking of bugs using the bug score. Thus, it can be concluded that the proposed approach can rank developers and bugs successfully.

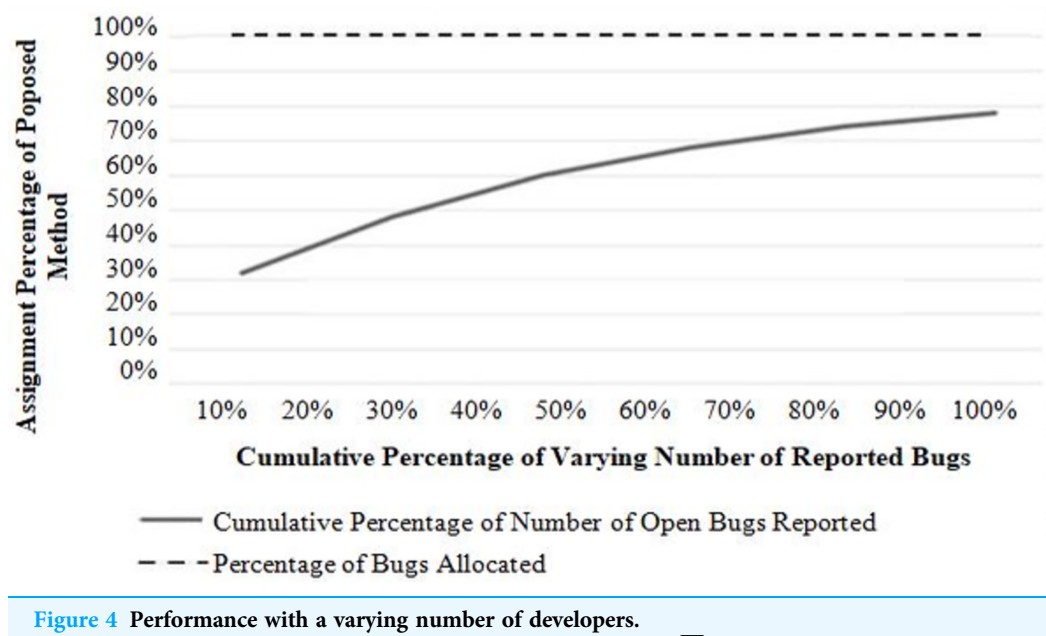

**Figure 4** Performance with a varying number of developers.

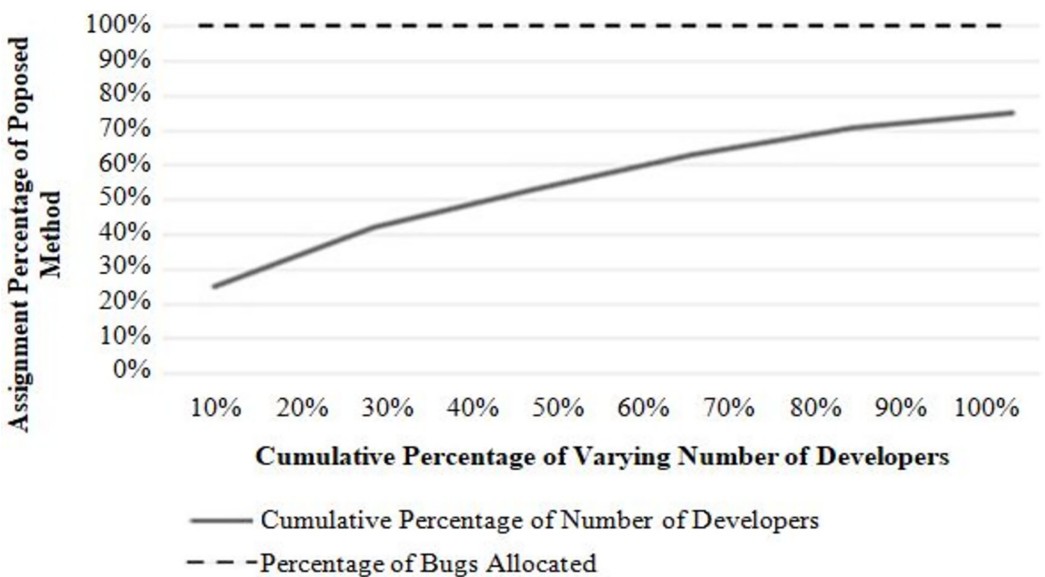

**Figure 5** Performance with the varying number of a bug report to be assigned.

## Assignment rate (RQ2)

This experiment measures the algorithm's effectiveness for a successful bug assignment to the potential developer. Two types of result analysis are presented: (i) performances with a varying number of developers and (ii) performances with a varying number of bug reports to be assigned. Figures 4 and 5 show the results of each analysis. In Fig. 4, each point on the graph represents a bug report. The x-axis represents the number of bug reports, and the y-axis represents the number of developers assigned to each bug report. The graph shows

**Table 7 Comparison with other state-of-the-art approaches.**

| Existing approach | Dataset used | Performance |
|---|---|---|
| *Almhana, Kessentini & Mkaouer (2021)* | Eclipse, Mozilla | Accuracy 77.43% (Eclipse), 77.87% (Mozilla) |
| *Jonsson et al. (2016)* | Eclipse | Accuracy 92.99% |
| *Zhang et al. (2013)* | Eclipse, Firefox | Precision 60%, recall 3% (Eclipse), Precision 51%, recall 24% (Firefox) |
| *Guo et al. (2020)* | Eclipse, Mozilla | Accuracy 53.10% (Eclipse), 56.98% (Mozilla) |
| *Alazzam et al. (2020)* | JDT-Debug, Firefox | Accuracy 89.41% (JDT-Debug), 59.76% (Firefox) |
| *Chen, Wang & Liu (2011)* | Eclipse, Mozilla | Accuracy 84.45% (Eclipse), 55.56% (Mozilla) |
| *Kashiwa & Ohira (2020)* | Eclipse, Mozilla | Accuracy 60.40% (Eclipse), 46.46% (Mozilla) |
| *Bhattacharya, Neamtiu & Shelton (2012)* | Industry | Accuracy 89% |
| *Jeong, Kim & Zimmermann (2009)* | Eclipse, Mozilla, Ant, TomCat6 | MRP 0.28 (Eclipse), 0.28 (Mozilla), 0.35 (Ant), 0.35 (TomCat6) MAP 56.42 (Eclipse), 44.49 (Mozilla), 36.48 (Ant), 36.54 (TomCat6) |
| *Guo et al. (2018)* | Eclipse | Reduced data set's rate-3.96% greater than the original data set's rate. |
| *Kashiwa (2019)* | Eclipse, Mozilla, and GNOME. | Analyzing the severity of bug reports should perform better. |
| *Kukkar et al. (2023)* | Eclipse, GCC, and Mozilla | Accuracy attained 81.7% |
| *Pan et al. (2022)* | Mozilla | Accuracy 92.99% |
| *Kashiwa & Ohira (2020)* | Eclipse, GCC, and Mozilla | Bug fix duration reduced by 35–41% |
| *Gupta, Inácio & Freire (2021)* | GitHub | Harmonic mean of precision 92.05, recall 89.21%, f-measure 85.09%, and accuracy attained 93.11% |
| *Gupta & Freire (2021)* | Ethereum | Reduced cost and time of bug fixing. |
| *Kukkar et al. (2023)* | Eclipse, Firefox, OpenFOAM, Mozilla | Improved by an average of 4%, 10%, and 12% |
| **Proposed approach** | Mozilla, Eclipse, NetBeans, Jira, OpenStack | Accuracy 92.42% (Mozilla), 92.72% (Eclipse), 90.38% (NetBeans), 91.34% (Jira), 89.38% (Freedesktop). Average accuracy 91.25%, Average precision 92.05%, Average Recall 89.04%, Average f-measure 90.05% and Average BTL 87.51% |

how the number of developers assigned changes as the number of bug reports increases. It can be seen that the successful allocation rate of the proposed approach is always 1 (100%). It shows that the proposed approach can effectively allocate bugs to various developers with the lowest cost when the number of developers changes. In this experiment, the number of bugs is fixed, and the performance is evaluated by increasing the number of developers with a regular increment of 5.

Referring to Fig. 5, bugs to developers can be allocated effectively with the lowest cost when the number of bugs changes. In this experiment, the number of developers is fixed, and the performance is evaluated by increasing the number of bug reports with a regular increment of 4.

## Comparison with other state-of-the-art approaches (RQ3)

A comparison study was conducted between the proposed approach and other existing state-of-the-art approaches. The theoretical result analysis focuses mostly on the accuracy parameter (given in Table 7) and is presented chronologically. The proposed work's

**Table 8 Summary of descriptive statistical reliability analysis of survey results.**

| Variable | Mean | Standard deviation | Cranach's alpha (α) |
|---|---|---|---|
| a) Is the proposed solution good enough for bug assignments? Does it make sense, in general? | 2.43 | 1.30 | 0.940 |
| b) Are there any constraints or dependencies which make solution infeasible and to what extend? | 3.13 | 1.21 | |
| c) Is it costly to use this solution in practice in terms of long process, fetching data, processing data *etc*.? | 3.02 | 1.24 | |
| d) Can we effectively automate the bug assignment process using proposed solution by taking explicit inputs (mentioned in proposed solution)? | 2.69 | 1.27 | |
| e) Do think more data/information should be considered without which the use of the proposed solution is difficult? | 3.02 | 1.24 | |
| f) If you change your input during execution - how the proposed solution affects your choice? Do you think it's a positive thing to incorporate? | 2.69 | 1.27 | |
| g) In comparison to existing practices followed in your organization, do you think this approach will make a difference in the bug assignment? | 3.13 | 1.33 | |

performance is evaluated against the results of the fuzzy logic-based Bugzie model (*Tamrawi et al., 2011*). Overall, it is clear that the performance measure of the proposed method is significantly higher than the Bugzie model (*Tamrawi et al., 2011*), with an average accuracy of 91.25%. The comprehensive evaluation of the proposed approach with five open-source projects confirms that effective bug triaging can be performed if both developers and bug ranks are considered. The system's overall accuracy for all datasets and processes is around $90 \pm 2\%$. Results indicate that using the proposed method can achieve high triage accuracy and reduced bug tossing length effectively.

## User study (RQ2)

To answer RQ2, a survey was conducted with 54 industry experts with 10 to 35 years of experience. 65% of these experts were males, 60% were between 30 and 45, and 30% were between 46 and 58. 5% were under 26 years old, and 5% were elderly (55 years and over). Only decision-makers, such as project managers, were involved. Responses on 55-point Likert scale were recorded for seven questions to measure the respondent's score. Likert is an ordered scale from which respondents can choose one option that best presents their view. A total of five represents strongly agree, whereas one represents strongly disagree. An additional "do not know" option was added to reduce the noise in the response data. The survey questions were directly mapped to the objective of the presented approach, answering the question, "*Can the proposed approach effectively perform bug assignment quantitatively and without ambiguity*?" Specifically, the following questions were formulated to capture the response.

Table 8 presents the descriptive statistical reliability analysis of survey results. It can be seen that there is adequate internal consistency with Cranach's alpha of the overall scale as 0.980. It indicates that each response is significantly correlated with the other. A value greater than 0.70 is considered suitable for the acceptance of the reliability of results. The key takeaways are as follows: (a) the existing methods tend to recommend bugs to only a few developers in comparison to the proposed approach that distributes the bugs among

developers such that at least one developer is recommended; (b) Assignments are made to make sure that each developer is able to fix the bug before the next release; (c) Unlike existing methods, our methodology considers many aspects such as developer expertise, performance, and bug value.

## CONCLUSION, LIMITATIONS, AND FUTURE SCOPE

This article presents a novel multi-criteria automated bug assignment approach to improve the quality of bug assignments in large software projects. It is a two-fold process that uses an evolutionary algorithm and intuitionistic fuzzy logic in a novel way to address trade-offs of supporting decision-making. It builds metrics for computing the developer's capability and metrics for the relative importance score of the bug. Besides this, an incentive mechanism for the developer's motivation is provided. Meta-features gather decisive and explicit knowledge about the developer's performance profile and bug importance from bug reports. Results of experiment evaluation on five open-source projects (Mozilla, Eclipse, NetBeans, Jira, and Free desktop) demonstrate that the proposed approach outperforms other approaches and achieves the harmonic mean of precision, recall, f-measure, BTL reduction, and accuracy of 92.05%, 89.04%, 90.05%, 87.51%, and 91.25% respectively with overall system accuracy of around 90 ± 2%. With the proposed approach maximization of the throughput of the bug, an assignment can be achieved effectively with the lowest cost when the number of developers or several bugs changes. Thus, the presented approach is simple, easy to use, and yet powerful in improving (a) the bug assignment process and (b) handling uncertainty and vagueness of expert judgment by creating a balance between multiple selection and assignment criteria using an evolutionary algorithm and intuitionistic fuzzy logic with reduced overhead in cost and time of bug fixing.

The proposed solution handles two significant issues (i) differentiating active and inactive developers and confusion over the assignment of bugs and (ii) identification of availability of developers according to their workload. Active developers are the ones who frequently and actively participate in the bug-triaging process. In contrast, inactive developers have not contributed to bug resolution for a long time by reducing bug-fixing delays and preventing re-assignment problems. One of the limitations of the proposed approach is that it does not handle the load balancing of developers in bug triaging. Although it determines the available developer, it does not distribute it evenly among available developers. In the future, we would like to address this issue.

### Funding

The authors received no funding for this work.

### Competing Interests

Varun Gupta is an Academic Editor for PeerJ Computer Science

## Author Contributions

- Chetna Gupta conceived and designed the experiments, performed the experiments, analyzed the data, performed the computation work, prepared figures and/or tables, authored or reviewed drafts of the article, and approved the final draft.
- Varun Gupta conceived and designed the experiments, performed the experiments, analyzed the data, performed the computation work, prepared figures and/or tables, authored or reviewed drafts of the article, and approved the final draft.

## Data Availability

The code is available in the Supplemental File.

- The Bug Report Assignment dataset is available at GitHub: https://github.com/anonymous-programmers/BugReportAssignment.

- YUI 3: The Yahoo User Interface Library is available at GitHub: http://github.com/yui/yui3

- The Julia Programming Language is available at GitHub: http://github.com/julialang/Julia

- Elasticsearch is available at GitHub: http://github.com/elastic/elasticsearch

- Salt is available at GitHub: http://github.com/saltstack/salt.

- Rails is available at GitHub: https://github.com/rails/rails.

## Supplemental Information

Supplemental information for this article can be found online at http://dx.doi.org/10.7717/peerj-cs.2111#supplemental-information.

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
