# Peer review of "Enhancing bug allocation in software development: a multi-criteria approach using fuzzy logic and evolutionary algorithms"

_PeerJ Computer Science, doi:10.7717/peerj-cs.2111_

## Round 0.1 · original submission · Major Revisions

Dear authors,

Thank you for submitting your paper. The reviewers have now commented on your article and suggested major revisions. When submitting the revised version of your article, it is better to clearly address all the reviews, comments and suggestions in terms of basic reporting, experimental design, validity of results and other comments.

Best wishes,

**Language Note:** PeerJ staff have identified that the English language needs to be improved. When you prepare your next revision, please either (i) have a colleague who is proficient in English and familiar with the subject matter review your manuscript, or (ii) contact a professional editing service to review your manuscript. PeerJ can provide language editing services - you can contact us at [email protected] for pricing (be sure to provide your manuscript number and title). – PeerJ Staff

Reviewer 1 ·

Basic reporting

1-It is mentioned by the authors that a new method is proposed to use fuzzy logic and evolutionary algorithms together to increase the error detection quality. However, more detailed information about how to detect this error is not given in the introduction of the article.
2- The literature study is not given critically, it seems that a raw literature review has been made. Reflect critical current literature studies into the article.
3- The article mentions that a multi-purpose evolutionary algorithm is used. Explain in more detail what these purposes are.
4-Why was PSO used as an evolutionary algorithm? Why was PSO chosen among many meta-heuristic methods?

Experimental design

5-I request the results to be compared using the Ant Colony Algorithm or the artificial bee colony algorithm. It is not correct to obtain results using a single meta-heuristic method.

Validity of the findings

6-It is understood that among the literature studies given in Table 7, there are no articles between 2016 and 2020, and there are also no articles between 2020 and today's 2024. The literature needs to be critically re-evaluated.
7-Figure 1 should be redrawn with high resolution.

Reviewer 2 ·

Basic reporting

This paper presents a novel hybrid algorithm combining multi-criteria fuzzy logic and a multi-objective evolutionary approach to automate bug management in bug tracking systems. It addresses the subjective assessment of bug attributes like severity and priority, typically based on user or developer intuition. The method enhances decision-making by:
Acquiring detailed and clear information about bug reports, factoring in developer workload and bug priority.
Developing metrics to evaluate developer capabilities, considering their expertise, performance, and availability.
Creating metrics to determine the relative importance of bugs.
The effectiveness of this approach was tested on five open-source projects - Mozilla, Eclipse, NetBeans, Jira, and Free desktop, demonstrating about a 20% improvement over existing methods.

Advantages:
1.This paper thoroughly reviews previous studies and identifies their shortcomings. It aims to bridge these gaps by offering potential solutions.
2. This paper is clearly written and easy to understand.

Disadvantages:
Experiments design: There are several issues in the experimental design of this paper. I will detail these issues in the following sections.

Experimental design

a. In the introduction section, the authors discuss the drawbacks of various techniques such as machine learning, information retrieval, and social network analysis, particularly in terms of cost and performance. To better demonstrate the superiority of their proposed method over these existing approaches, the authors should consider incorporating metrics such as computational cost into their comparison.

b. Baseline selection: the paper primarily compares with studies that are about a decade old, as noted in Table 7. Additionally, the performance evaluation does not appear to cover all five datasets comprehensively. (e.g. Tamrawi et al., 2011 [13] uses only one dataset: Eclipse.) For a more robust and fair comparison, it is advisable for the authors to include more recent studies as baselines. They should also ensure uniform experimental conditions, such as using the same datasets, server, or device settings, across all comparisons.

Validity of the findings

Upon conducting the suggested experiments in section 2 regarding Experimental Design, it is imperative for the authors to meticulously document the experimental procedure and thoroughly analyze the results obtained. This documentation should encompass a detailed description of the experimental setup, including the specific parameters, datasets used, and the computational environment. Additionally, the analysis should delve into interpreting the results, highlighting key findings and drawing comparisons with baseline studies. The authors could also discuss any unexpected patterns observed during the experiments, or situations where the prior works shows better results, and provide a reasonable explanation for these occurrences. A comprehensive analysis is crucial to substantiate the effectiveness and reliability of the proposed method, thereby validating the research findings.

Additional comments

Upon conducting the suggested experiments in section 3 regarding Experimental Design, it is recommended for the authors to meticulously document the experimental procedure and thoroughly analyze the results obtained. For the selected baselines, it would also be better to include a detailed description of the experimental setup, including the specific parameters, datasets used, and the computational environment. Additionally, the analysis should delve into interpreting the results, discussing the results in detail, and highlighting key findings.

---

## Round 0.2 · Minor Revisions

Dear authors,

Thank you for your submission. Your article has not been recommended for publication in its current form. However, we do encourage you to address the concerns and criticisms of the reviewers and resubmit your article once you have updated it accordingly.

Best wishes,

Reviewer 1 ·

Basic reporting

The authors reflected everything I wanted in the previous revision to the article.

Experimental design

The authors reflected everything I wanted in the previous revision to the article.

Validity of the findings

The authors reflected everything I wanted in the previous revision to the article.

Additional comments

The authors reflected everything I wanted in the previous revision to the article.

·

Basic reporting

The manuscript proposes an algorithm combining multi-criteria fuzzy logic and a multi-objective evolutionary approach to automate bug management in bug-tracking systems. It calculates a developer's capability score based on several aspects and uses IFS to calculate the bug's importance score. Then it uses particle swarm optimization to assign bugs to developers. The algorithm is novel.

However, some things need improvement.

1. Some concepts are used without any introduction. For example, it is hard to follow the section "Bug Value Computation". You should first briefly introduce IFS and then mention how it works in your scenario. Starting from line 368, some symbols are used without definition.

2. Some sentences are hard to understand. For example, lines 286-290, 315-317, 356-358 and 538-539. The logic is broken in these sentences.

Experimental design

How do you acquire the ground truth ranking of a bug or a developer? What is considered a successful match or an unsuccessful match? These are not mentioned in the setting.

Validity of the findings

1. It is hard to distinguish the curves in your figure. You should use different colors or some dashed curves to draw your figures.

2. In RQ2, I can not understand Figures 4 and 5. Should figure 4 and 5 be exchanged? In Figure 4, the x-axis should be the number of developers but your use bugs. In Figure 5, the x-axis should be bugs but you report developers. In addition, what does the y-axis mean in each figure?

3. In RQ3, the performance column is hard to interpret. Words like "improved by an average of 4%, 10%, and 12%" and "Reduced cost and time of bug fixing" are confusing. You should list the accuracy of each approach working on each dataset and then make the comparison.

4. In RQ4, How do you experiment and what do you measure? How can I interpret the table results?

---

## Round 0.3 · Minor Revisions

Dear authors,

Thank you for submitting the revised article. Feedback from the reviewer after revision is now available. It is not recommended that your article be published in its current format. However, we strongly recommend that you address the minor issues raised by the reviewer.

Best wishes,

·

Basic reporting

Compared to the previous submission, the revision gives more explanation of the concept and fixes the broken sentences. I am satisfied with the revision.

Experimental design

The new text addresses my concern.

Validity of the findings

Most problems are fixed but I still have concerns about RQ4.

In RQ4, the table is still too hard to interpret. For example, in Table 8, "Developer-Rank-Error vs Fig. 3(a) accuracy", I can not find Figure 3(a) in the manuscript. Please fix it.

In addition, the author should list two rows of data for "Developer-Rank-Error vs. Fig. 3(a) accuracy", including data before error injection and after error injection instead of a single row. Please also add the necessary text to describe the table in your section.

---

## Round 0.4 · accepted · Accept

Dear authors,

Thank you for clearly addressing all the reviewers' comments. I confirm that the quality of your paper is improved. The paper now seems to be ready for publication in light of this last revision. When submitting the final version for production step, please do the following:

1. Pay special attention for the usage of blank characters. Please correct “[6-13]match”, “technique [11,12]”, “Tran et al. [13]discovered”, “along with[14, 19, 28],”, “Fig.1 sketches”, “also, 0<i≤ n”, “µij& υij”, “equation (18)obtained above”, “Table 2.Weights”, “ps = (p1,p2,p3…pn),”, “Table 5.Statistics of Filtered Data”, “Table6.Top-1 to Top-10 Accuracies of Various Classifiers”, “Table 7.Comparison with Other State-of-the-Art Approaches”, and etc.

2. For multiplication operator, please use a single sign. Not “*”, “x”, “.” and etc. together.

3. Write the variables in italics. See for example lines 348-350, 393, 394, 3983 506, 508, 641, 642, 651, 656, 678, 679, and etc.


Best wishes,